# Devices and Treatments to Address Low Adherence in Glaucoma Patients: A Narrative Review

**DOI:** 10.3390/jcm12010151

**Published:** 2022-12-24

**Authors:** Barbara Cvenkel, Miriam Kolko

**Affiliations:** 1Department of Ophthalmology, University Medical Centre Ljubljana, 1000 Ljubljana, Slovenia; 2Medical Faculty, University of Ljubljana, 1000 Ljubljana, Slovenia; 3Department of Drug Design and Pharmacology, University of Copenhagen, 2100 Copenhagen, Denmark; 4Department of Ophthalmology, Copenhagen University Hospital, Rigshospitalet, 2600 Glostrup, Denmark

**Keywords:** adherence, drug delivery system, glaucoma, laser trabeculoplasty, medical treatment, minimally invasive glaucoma surgery

## Abstract

Poor adherence to topical glaucoma medications has been linked to worse visual field outcomes in glaucoma patients. Therefore, identifying and overcoming the adherence barriers are expected to slow down the progression of disease. The most common barriers to adherence, in addition to the lack of knowledge, include forgetfulness, side effects of medications, difficulties with drop instillation and low self-efficacy. Symptoms and signs of ocular surface disease, which importantly reduce patients’ quality of life, are decreased by using preservative-free topical medications. Sustained drug delivery systems using different vehicles seem promising for relieving the burden of drop administration. Currently, only the bimatoprost sustained-release intracameral implant is available for clinical use and single administration. In the era of digitalization, smart drug delivery-connected devices may aid adherence and, by sharing data with care providers, improve monitoring and adjusting treatment. Selective laser trabeculoplasty as first-line treatment delays the need for drops, whereas minimally invasive glaucoma procedures with and without devices combined with cataract surgery increase the likelihood of patients with early-to-moderate glaucoma to remain drop free or reduce the number of drops needed to control intraocular pressure. The aim of this narrative review is to present and discuss devices and treatments that may improve adherence by reducing the need for drops and side effects of medications and aiding in glaucoma monitoring. For the future, there is a need for studies focusing on clinically important outcomes, quality of life and the cost of intervention with longer post-interventional follow up.

## 1. Introduction

Lowering of intraocular pressure (IOP) is the only proven treatment to slow or delay the progression of glaucoma [1,2,3,4]. Medical treatment is the most common approach to achieving an individual eye “safe” IOP, followed by monitoring to determine the rate of progression [5]. As glaucoma is chronic optic neuropathy, and patients usually need to take lifelong medications. Therefore, adherence to a treatment regimen is crucial to maintain visual function. The reported rates of nonadherence to topical glaucoma medication vary widely from 16% to 67%, reflecting different methods to identify nonadherence as well as absence of a quantitative standard for measuring adherence to glaucoma medication [6]. Adherence over a longer period has been found to be even lower. Thus, only one-quarter of patients with newly diagnosed glaucoma continued their glaucoma medication after 2 years of follow up in Taiwan [7], whereas only 15% with newly diagnosed glaucoma in another study showed persistently good adherence over 4 years of follow up [8]. For most patients of newly prescribed glaucoma medications, adherence patterns observed in the first year of treatment mirror adherence patterns over the subsequent 3 years [8]. It is recognized that poorer adherence to glaucoma treatment leads to higher IOP, greater fluctuations in IOP and consequently progression of glaucoma [9,10,11]. Glaucoma patients who reported less than 80% adherence to their prescribed medications were significantly more likely to have worse visual field defects [11]. Few studies longitudinally assessed the relationship between medication adherence and visual field progression. A longitudinal study assessing adherence in 35 glaucoma patients reported that patients with a stable visual field had a significantly higher median adherence rate of 85% compared to progressing patients with a median medication adherence of 21% [12]. In another study, patients randomized to the treatment arm of the Collaborative Initial Glaucoma Treatment Study were assessed for medication adherence using telephone interviews scheduled at the same 6-month intervals as clinical visits but on different days and followed up for an average of 7.3 years [13]. Scheduling timing of telephone interviews independently of clinical visit represents a strength of the study as nonadherence is often not admitted in front of the treating physician. Patients who reported never missing a dose of medication over the follow up had an average mean deviation (MD) loss of 0.62 dB over time, consistent with age-related loss, whereas patients missing medication doses at one-third and two-thirds of visits had an average loss of 1.42 and 2.23 dB of MD, respectively [13]. These findings indicate a dose–response relationship between medication adherence and visual field progression. A range of factors affect adherence and persistence, with one study identifying 71 barriers to adherence over four categories: situation factors, medication regimen, individual patient factors and medical provider issues [14]. Patients with poor adherence cited several barriers to medication adherence, which varied between individuals. The most important barriers associated with nonadherence included forgetfulness, low self-efficacy, difficulties with drop instillation and treatment schedule, side effects of medication, lack of motivation, poor education and other specific individual and age differences [15,16,17]. In addition, certain types of disabilities such as having a limb disability, being in a vegetative state, and having dementia reduce glaucoma medication adherence by up to 17.6% [18]. Therefore, approaches addressing adherence to glaucoma medications need to be multifaceted and individually tailored [19].

The purpose of this narrative review is to discuss strategies that may improve adherence by reducing the side effects of drops, the number of instillations and the number of medications required to control intraocular pressure. These approaches include medical treatments using preservative-free drops, intracameral sustained drug delivery and various drug delivery systems still undergoing clinical trials, smart drug delivery-connected devices as well as selective laser trabeculoplasty and minimally invasive glaucoma surgery.

## 2. Materials and Methods

We conducted a literature search using the PubMed database. The following search terms were included “adherence” AND “glaucoma medication”, “topical treatment” AND “glaucoma”, drug delivery systems and glaucoma treatment, “selective laser trabeculoplasty”, and “minimally invasive glaucoma surgery”. The articles retrieved were reviewed for their title, abstract, and language. Articles included were in in English language published before August 2022 including clinical trials in humans, editorials, reviews, systematic reviews, and meta-analyses. We focused on randomized clinical trials (when available). After retrieving relevant articles using keywords, a search was performed through the reference lists of the chosen studies and additional papers were selected.

## 3. Medical Treatments

### 3.1. Preservative-Free Drops

Adverse drug reactions are a major barrier to adherence and persistence. Local side effects can vary from minor dry eye symptoms to allergic and toxic-inflammatory responses [20]. The use of preservatives, especially benzalkonium chloride (BAK), is a known cause of ocular surface disease (OSD) in patients taking topical IOP-lowering eye drops [21]. In a survey study in French glaucoma patients, 62% of the patients cited at least one OSD side effect and 19% of patients at least four such side effects [22]. The reported prevalence of OSD among glaucoma patients in other studies is similar, between 60% and 70%, which is much higher than in age-matched subjects without glaucoma (between 15% and 33%) [23,24,25]. Self-reported nonadherence over 9.4 years, defined as missing ≥5% of the prescribed eye drops, was reported by 30% of participants [26]. Individuals who experienced side effects reported significantly higher levels of nonadherence than those who did not (37.6% vs. 18.4%) [26]. The side effects of OSD were associated with a reduced quality of life and worsening of quality-of-life scores correlated with reduced adherence captured by a questionnaire [22]. Several studies reported that the severity and prevalence of OSD in glaucoma patients correlated with the number of preserved drops per day and duration of treatment [24,27,28].

The improvement of symptoms and signs of OSD after switching from preserved to preservative-free eye drops has been shown in many studies [29,30,31,32,33]. In a large prospective survey including 4107 participants, Pisella et al. reported that patients taking preserved eye drops had significantly more symptoms and signs of OSD than those taking preservative-free eye drops [29]. For patients experiencing more pronounced signs and symptoms of ocular irritation, a treatment change from preserved to preservative-free eye drops significantly decreased the prevalence of all symptoms and signs. Jaenen et al. in a cross-sectional study including 9658 patients assessed subjective symptoms and signs of ocular irritation [30]. Each symptom and all the signs (blepharitis, eczema, hyperemia, and fluorescein staining) were significantly more frequent in patients taking preserved than in patients on preservative-free eye drops [30]. At the time of these two studies, most patients used preserved hypotensive drops, and the choice of preservative-free formulations on the market was limited. Following the launch of preservative-free tafluprost and a preservative-free fixed combination of timolol with dorzolamide, preservative-free latanoprost and bimatoprost and later their fixed combinations with timolol became available. Several studies have compared the efficacy and tolerability of preservative-free prostaglandin analogues and their fixed combinations to their preserved counterpart and found that preservative-free formulations are noninferior in their IOP-lowering effect and associated with less signs and symptoms of OSD. Patients with ocular symptoms or signs of OSD on preserved latanoprost (Xalatan^®^; Pfizer, New York, NY, USA) were switched to preservative-free tafluprost (Taflotan^®^; Santen Oy, Finland) which had similar IOP-lowering effect as preserved latanoprost but was better tolerated and resulted in a decrease in symptoms and signs, and improved quality of life and patients’ satisfaction [31,34]. The same efficacy and better tolerability have been shown for a preservative-free latanoprost (Monopost^®^, Thea Pharmaceuticals, France) compared with preserved latanoprost (Xalatan^®^; Pfizer, New York, NY, USA) [35]. Pillunat et al. [33] evaluated in an open-label study efficacy and tolerability of preservative-free fixed combination of tafluprost/timolol (Taptiqom^®^; Santen Oy, Tampere, Finland) in 1157 patients. Preservative-free fixed combination lowered IOP in all subgroups of patients: treatment naïve, prior monotherapy and prior fixed combinations. At the final visit at 16 weeks, symptoms, and signs of OSD improved in patients with prior medical therapy and, using a simple questionnaire, 90% of patients rated treatment comfort as very good or good [33].

### 3.2. Sustained Drug Delivery Systems

Development of sustained drug delivery vehicles has been an ongoing search to improve adherence among patients. These drug delivery systems may be applied onto the ocular surface (contact lenses, nanoparticles, microspheres, extraocular inserts), in the puncta (punctal plugs) or injected into the eye. Different novel drug delivery techniques are in different stages of development and only one, the intracameral bimatoprost SR ocular implant (Durysta^®^; Allergan, Irvine, CA, USA) has been recently approved by the FDA for sustained IOP reduction [36].

#### 3.2.1. Nanoparticles

To overcome the limitations of topical antiglaucoma medications, nanoparticulate (NP) delivery systems may improve solubility of the drug and corneal penetration, increase concentration at the target tissue, reduce irritation and systemic side effects and provide dose accuracy and sustained release of the drug [37]. Nanoparticles, tiny structures ranging from 1 to 100 nm in size, can bypass biological barriers and deliver drug to the target tissue. Nanoparticles are used in different shapes such as nanoemulsions, dendrimers, liposomes, nanospheres, hydrogels, nanocrystals, nanodiamonds, microspheres, niosomes, nanofibers, and nanocapsules [37]. Lipid and polymer nanoparticles are usually used to carry the drugs, isolate their contents from degradation and regulate their release. Several drugs using a NP delivery system are under investigation, but currently none has been approved for clinical use [38].

#### 3.2.2. Contact Lenses

The contact lens drug delivery system is also very attractive. As the bioavailability of drugs with drop instillations is very low, the incorporation of drugs into the contact lens matrix increases the drug residence time on the cornea and improves drug bioavailability by more than 50% compared to eye drops [39]. Achieving sustained or prolonged release of the drug from the contact lens allows for reduced frequency of drop instillation and potentially improved adherence. Different drug loading methods are used to incorporate drugs into polymeric support [39,40]. At present, most of the glaucoma drug-loaded therapeutic contact lenses are in the preclinical or clinical stages and data regarding safety, efficacy and pharmacokinetics are required [41].

#### 3.2.3. Extraocular Inserts

Ocular inserts of different forms and sizes are shaped to fit into the conjunctival fornices. These inserts increase the ocular surface contact time of the drug, improve its bioavailability and reduce the need for frequent drop instillation. Among the first approved ocular inserts, Ocusert™ was placed under the eyelid and released pilocarpine over one week [42]. Although it was effective in reducing IOP, its use was limited by device dislodgement and high cost. Ocusert™ is not available on the market since 1993.

Another insert, bimatoprost sustained-release fornix ring-type insert, is in the late stage of development for clinical use. The insert achieved IOP reduction over 6 months similar to timolol 0.5% BID drops, was safe and well tolerated [43]. In a 13 months open-label extension study the bimatoprost ring showed good retention rate with a median IOP reduction of 4 mmHg (interquartile range 2–6) [44]. The most frequently reported adverse events from both studies were mucous eye discharge (16%), conjunctival hyperemia (14%) and punctate keratitis (12%).

#### 3.2.4. Punctal Delivery Systems

Different solutions, suspensions, emulsions, nanoparticle or microparticle or liposome suspensions can be loaded into the core of the plug [45]. The latanoprost punctal plug delivery system (Mati Therapeutics) was loaded with 70.5 µg of latanoprost per device. When two such plugs were inserted in the upper and lower puncta, the mean IOP was reduced by 5.7 mmHg (22.3%) from baseline after 4 weeks [46]. The latanoprost-loaded punctal plug was well tolerated, with tearing reported as the most frequent adverse event.

The travoprost punctal plug (OTX-TP, Ocular Therapeutics, Bedford, MA, USA) is a rod-shaped hydrogel rod that swells in the canalliculus, thus preventing extrusion. Travoprost is encapsulated in polylactic acid microparticles for sustained release to the tear film over 90 days [47]. In the double-masked phase 2b study (NCT02312544) comparing the safety and efficacy of sustained-release travoprost plug delivery to timolol eyedrops in patients with open-angle glaucoma or ocular hypertension, OTX-TP plugs reduced IOP by 4.5–5.7 mmHg, whereas timolol reduced IOP by 6.4 mmHg–7.6 mmHg. The superb efficacy of timolol eye drops is likely the effect of decreased wash-out through the nasolacrimal ducts by inert punctal plugs. In an Asian population, sustained-release travoprost reduced IOP by 24% at 10 days, and by 15.6% at day 30 [48].

Among the major limitations of the punctal drug delivery system is that only low drug doses, typically required for potent drugs (e.g., prostaglandins and corticosteroids), can be embedded into the plug core matrix. At present the only punctal delivery system available on the market is dexamethasone 0.4 mg insert (Dextenza™, Ocular Therapeutix) approved by the FDA in 2018 for the treatment of ocular inflammation and pain following ophthalmic surgery.

#### 3.2.5. The Periocular Drug Delivery System

For the subconjunctival route, several delivery systems can be used such as implants, microspheres, nanospheres, liposomes, and gels [45]. Most of the studies were performed in rabbits by injecting a subconjunctival formulation of timolol, brimonidine, latanoprost and carbonic anhydrase inhibitors and achieved good IOP reduction without signs of inflammation for up to 90 days, depending on the delivery system used [49,50,51,52,53,54]. In a pilot study (NCT01987323) including six patients, a nanoliposome-based latanoprost delivery system was well tolerated, and in five out of six patients IOP reduction achieved at 3 months was as effective as previous reports of latanoprost ophthalmic solution [55]. A recently completed randomized trial including 80 participants (NCT02466399) comparing the efficacy of subconjunctival liposomal latanoprost injection to latanoprost ophthalmic solution reported a mean change in IOP at month 3 of −2.3 mmHg (SD 4.6) and of −6.4 mmHg (SD 2.9), respectively. Adverse events were reported in the liposomal latanoprost group only, with the most frequent being conjunctival hemorrhage (26.4%), foreign body sensation (17.0%), and conjunctival hyperemia (13.2%). To date, no subconjunctival delivery system has been approved, suggesting inherent delivery and efficiency limitations associated with these delivery systems.

#### 3.2.6. Intraocular/Intracameral Drug Delivery

Bimatoprost intracameral implant 10 µg (Durysta^®^; Allergan, Irvine, CA, USA) is the only sustained-release glaucoma therapy approved by the FDA in March 2020 for the lowering of IOP in patients with open-angle glaucoma or ocular hypertension [56]. It is a rod-shaped biodegradable implant based on a poly (lactic-co-glycol) acid matrix Novadur^®^ platform, used in dexamethasone intravitreal implant (Ozurdex^®^, Allergan, Irvine, CA, USA) and provides steady bimatoprost release for up to 6 months [57]. A single-use, 28-gauge sterile applicator is used for intracameral administration. Several doses were studied in clinical trials with the 10 µg of bimatoprost having the best balance between safety and efficacy [58,59,60]. The 10 µg of drug released is equivalent to a single drop of bimatoprost 0.03% ophthalmic solution [61]. The FDA approval for a new drug application was based on the results of two phase 3 clinical studies comparing administration of 10 and 15 µg bimatoprost implant with twice daily timolol maleate 0.5% eye drops. Both implants showed noninferiority to topical timolol eye drops in lowering IOP through 12 weeks but with more adverse events such as corneal edema and endothelial cell loss than in the topical timolol group, especially with the 15-µg implant after repeated administration [60,62]. Long-term retention of the implant beyond the optimal drug effect period is another disadvantage when considering readministration. For this reason, the 10 µg bimatoprost implant with a better benefit–risk ratio was approved by the FDA and limited to a single intracameral administration. Pooled analysis of the two phase 3 studies reported that the percentage of bimatoprost 10 µg-treated patients with at least 20% IOP lowering from baseline in the study eye was 72% at week 12 and 57% at week 15 [63]. The IOP-lowering efficacy of the bimatoprost 10 µg implant declined from week 12 to week 15. In the 24-month phase 1/2 study, 21 patients received the 10 µg bimatoprost implant in the study eye and topical bimatoprost 0.03% in the fellow eye [59]. The percentage of eyes receiving the bimatoprost implant with at least 20% reduction from baseline was 76.2% (16/21) at week 12 and 52.4% (11/21) at week 16. Interestingly, in 5 patients who reached month 24 without re-treatment or additional hypotensive medication, the IOP-lowering effect at the final visit was similar to the effect of once-daily topical bimatoprost [59].

Ongoing clinical trials (NCT03891446, NCT03850782) are evaluating the efficacy of IOP lowering and the safety of readministration of the bimatoprost implant (Durysta^®^) over 24 to 48 months. Of interest are clinical trials comparing the efficacy and safety of bimatoprost SR to selective laser trabeculoplasty (SLT), of which one (NCT023636946) was completed last year. This study included 144 participants with open-angle glaucoma or ocular hypertension who were not adequately managed with topical IOP-lowering medications for reasons other than medication efficacy (e.g., due to intolerance or nonadherence) and were randomized to the bimatoprost SR 15 µg or SLT treatment groups. The primary outcome measure was change from baseline IOP at week 4, 12 and 24. The bimatoprost SR 15 µg was noninferior in IOP reduction to SLT at all scheduled visits. The second ongoing trial (NCT02507687) is comparing the IOP-lowering effect and the safety of bimatoprost SR (Durysta^®^) compared with SLT in patients with open-angle glaucoma or ocular hypertension who are not adequately managed with topical IOP-lowering medication. The findings of both aforementioned ongoing studies can support clinical decision, but also the long-term safety, repeatability and cost-effectiveness need to be considered. As a result of the prolonged IOP-lowering effect of bimatoprost SR, an ongoing trial is also investigating the efficacy and safety of treat and extend (NCT03850782) of Durysta^®^. No current results are available. 

Another intracameral implant in a phase 2 clinical trial (NCT02371746) is the ENV515 travoprost extended release (XR) (Envisia therapeutics, Research Triangle Park, NC, USA), using a biodegradable polymer as the drug delivery system. Single intracameral administration of a low dose of the ENV515 reduced the mean IOP by 6.7 mmHg (SD 3.7) at 11 months. Lowering of IOP was comparable to latanoprost, bimatoprost (reports) and the in-study 0.5 timolol maleate topical daily drops. Ocular hyperemia, punctate keratitis and foreign body sensation were the most common adverse events (NCT02371746). 

The iDose travoprost implant (Glaukos Inc., San Clemente, CA, USA) is a titanium intracameral delivery system that elutes travoprost. It is placed through a small corneal incision and anchored to trabecular meshwork. The membrane within the implant controls the release of travoprost into the anterior chamber. Once depleted, the implant can be removed and exchanged for continued treatment [47]. Ongoing clinical trials (NCT02754596, NCT03868124, and NCT03519386) are comparing different elution rates of the travoprost implant to topical timolol 0.5% dosed twice daily. In a phase 2b study (NCT02754596), mean IOP lowering at the 36 months was 8.3 and 8.5 mmHg in the fast and- slow-release travoprost implant, respectively, versus 8.2 mmHg in the timolol control arm. The 36-month phase 2 data did not show clinically significant corneal endothelial cell loss, no serious corneal adverse events and periorbital fat atrophy and conjunctival hyperemia in either elution arm [64]. Repeated procedures and presence of the implant in the angle for continued treatment may be associated with adverse events related to surgical procedure and long-term effect on the corneal endothelial cells. A phase 2 study is evaluating the safety of the operative and surgical exchange procedure of the travoprost intraocular implant (NCT04615403).

Travoprost for a slow and extended release, OTX-TIC (Ocular Therapeutix, Bedford, MA, USA), has also been incorporated in a fully biodegradable implant that is administered into the anterior chamber with a 27 G or 26 G needle. In OTX-TIC, travoprost-loaded microparticles are embedded in hydrogel, which allows for an extended release of travoprost for a 4–6-month duration. A phase 2 study (NCT05335122) is evaluating the efficacy and safety of the OTX-TIC low- and high-dose intracameral implant in patients with open-angle glaucoma or ocular hypertension and comparing 2 travoprost dose strength to a single injection of bimatoprost SR (Durysta^®^).

## 4. Monitoring Devices and Smart Drug Delivery Systems

The challenge of how to monitor adherence has been differently addressed and some studies used more than one method. The most common objective way of measuring adherence is electronic monitoring using a medication event monitoring system (MEMS), followed by pharmacy records [65]. A MEMS is a cap that fits on bottles and records the time and date each time the bottle is opened and closed. The Travatan Dosing Aid was among the first electronic monitoring devices designed to hold a bottle of travoprost. The device had attached base that recorded time when the lever that administered the medication was fully depressed, and the data were downloaded from the device [66]. Electronic monitoring using a MEMS has been used to assess the rate of adherence, identify patients’ characteristics associated with poor adherence and evaluate its change after different interventions to improve adherence [67,68,69,70,71,72,73]. Adherence data collected by a MEMS have shown that patients with glaucoma, especially those newly diagnosed were likely to overreport the percentage of doses taken [70,74,75]. Recently, Japanese researchers have developed and evaluated an eye dropper bottle sensor system comprising motion sensor with automatic motion waveform analysis using deep learning to accurately measure adherence of patients with antiglaucoma ophthalmic solution therapy [76]. An eye dropper bottle sensor was installed at patients’ homes, and they were asked to instill the medication and manually record each instillation time for 3 days. Waveform data were automatically collected from the eye dropper bottle sensor and judged as a complete instillation by the deep learning instillation assessment model. The eye dropper bottle sensor system successfully auto extracted the instillation data of 20 patients with glaucoma for 3 days with 100% accuracy in a moment and may be an option to objectively measure adherence in clinical practice [76].

Innovative solutions have been developed in the area of smart drug delivery. The benefits include better monitoring, user support, uploading data from different devices that are integrated using different platforms and made accessible to health care providers. Smart drug delivery makes treatment management for patients easier and may improve their adherence, integrated data shared with the eyecare providers can provide better monitoring and communications, all of which may increase treatment efficacy. In addition, smart drug delivery enhances treatment efficiency as the prescriptions are timely fulfilled without stopping therapy, which may lead to improved control of disease and less hospitalization. These innovations were introduced to improve adherence and monitoring of other chronic diseases, such as diabetes and asthma. Various smart insulin pens have been developed, including smart pen caps that automatically capture injection data and enable immediate transmission of data from the pen via Bluetooth or near-field communication to a smartphone application and into digital storage, so the stored data can be viewed by health care providers or caregivers [77,78,79]. Studies showed that patients preferred smart insulin pens as they increased their confidence in diabetes management, but there was a lack of published data regarding smart insulin pens with connectivity [77].

Kali Drop (Kali Care, Santa Clara, CA, USA) represents a potential improvement in the eye care by directly measuring regimen adherence in patients using topical medications [80]. This device is a compact, 3G wireless monitor that electronically transmits medication use (e.g., number of drops dispensed, time, and date taken) in real time through wireless networks to a user-friendly interface that may be used by patients, caregivers, and providers to view and track adherence to topical therapy. The device was used in a pilot study comparing use of topical medications for 2 months between a wireless monitoring device and validated self-reported measures of glaucoma medication adherence [81]. Median adherence as measured by the device and self-report differed and dropped slightly after 1 month for both. This suggests that despite participating in a study and knowing they will be monitored, adherence wanes over short time. The majority of subjects found the device easy to use and reported that it did not interfere with their daily activities, and they were not bothered by the physician tracking their eye administration. However, this study included a small sample (23 subjects) with a short follow up. Studies including more patients in a real-world setting with longer monitoring would add additional information about usability of this device.

E-Novelia^®^ (Nemera, La Verpillière, France) is a smart drug delivery-connected device. It is designed as an add-on to existing preservative-free formulations for glaucoma or dry eye treatment and is applicable across different medications. It has the potential to improve an already existing way of drop instillation and adherence [82]. This approach requires the use of the company’s own custom eye drop medication bottles to latch to a system that contains the embedded sensors. The device has several electronic features that could be transferred across multiple device platforms: tilt sensor and LED indication for device positioning, location tracking, remaining drug indicator, treatment history and compliance, shelf-life management, drop detection, electronic instructions for use, smartphone application and notification, shaking formulation indication, and RFID tag on eyedropper bottle to collect data.

In the United States, an intelligent sleeve, a monitoring system capable of detecting and quantifying eye drop medication use without altering the original medication packaging has been developed [83]. The prescription bottle is placed in the sleeve with the embedded sensors and electronics that measure fluid level, dropper orientation, the state of the dropper top (on/off), and rates of angular motion during an application. The sleeve was tested with ten patients (age ≥ 65) and successfully identified and timestamped 94% of use events [83]. Data from the sensors are transmitted from the system to a smart phone or another Bluetooth-connected device. Health care providers can use this information to support clinical decisions. 

This technology has the potential to be useful for patients, and health care providers that will benefit from following and adjusting treatment. The limitation of these devices is that they measure drop dispensing and not really drop instillation into the eye itself. A pilot study using imaging system to record video of the drop technique has shown that few patients were able to properly apply the drops. Most had issues either getting the drops in their eyes, applying the correct number of drops, touching the bottle to the eye or adnexa or some combination of the above [84]. We did not find any studies evaluating adherence in the real world using smart drug delivery-connected devices in glaucoma patients.

## 5. Selective Laser Trabeculoplasty

Selective laser trabeculoplasty (SLT) has fewer adverse events, improved repeatability and ease of use compared to its predecessor argon laser trabeculoplasty. It is an outpatient laser procedure which lowers IOP by increasing aqueous outflow through the trabecular meshwork. The procedure is indicated to lower IOP in open-angle glaucoma and ocular hypertension as first-line treatment or as an add-on or replacement treatment (e.g., nonadherence or intolerance) [5]. The LiGHT trial showed that there was no difference between SLT and eye drops used for first-line therapy over the 36 months period in achieving target IOP (20% or 30% IOP reduction), health-related quality of life, adverse events and treatment adherence in newly diagnosed patients with open-angle glaucoma and ocular hypertension [85]. This trial also reported that people who were given eye drops as first-line treatment, used more eye drops and required the use of more than 1 eye drop medication at 12 months, compared with people who were given SLT as first-line treatment. Furthermore, three-quarters of the patients initially treated with laser did not need any eye drops for the first 3 years of treatment and had a reduced need for surgery [86]. To achieve target IOP over 36 months, SLT needed to be repeated in approximately 15% of people in the SLT arm within the first year [86]. Visual field outcomes (median 9 visual fields over 48 months) showed that a slightly greater number of eyes (56 eyes (9.5%)) treated with medical therapy first had fast visual field progression defined as total deviation progression < −1 dB/year compared with those treated first with SLT (32 eyes (5.4%)) [87]. Cost-effectiveness analysis, pertinent to the United Kingdom, where the trial was conducted, showed that first-line treatment with 360° SLT was more effective and less costly compared with eye drops and should be offered as initial treatment in patients with open-angle glaucoma and ocular hypertension meeting the inclusion criteria of this trial [85]. No difference in adherence between the medical and SLT first-treated arm in the LiGHT trial may be due to patient selection, including highly motivated people with extensive support, which is not the case in practice [7,75,88]. Another prospective multicenter clinical trial comparing the effectiveness of SLT with topical medication as initial treatment did not find that SLT was superior over medication in improving glaucoma-specific health-related quality of life in newly diagnosed primary open-angle and exfoliative glaucoma patients over 2-year follow up [89]. More individuals in the medication arm had conjunctival hyperemia and eyelid erythema compared with the SLT at 24 months. Successful IOP reduction, defined as IOP-lowering of 25% or more from baseline, was superior in the medication arm compared with the SLT arm. The differences in findings between the two studies are probably caused by the differences in trial design, population and sample size. For participants with early to moderate primary open-angle and exfoliation glaucoma, no separate analysis by the subtype of glaucoma was performed [89,90].

Based on the evidence of the LiGHT trial regarding cost effectiveness, the National Institute for Health and Care Excellence (NICE) guidelines recommends that 360° SLT should be offered as first-line treatment to people with newly diagnosed ocular hypertension with IOP of 24 mmHg or more (and if they are at risk of visual impairment during their lifetime) or chronic open-angle glaucoma [91]. In the published literature, the most common adverse events are transient elevation of IOP, especially in eyes with heavily pigmented trabecular meshwork within the first 2 h after SLT (greater than 6 mmHg in 3.4% of eyes) and mild iritis which resolves in a few days [92,93]. Rare adverse events described in case reports include transient changes in the corneal endothelium on specular microscopy in eyes with pigment on endothelium [94], recurrence or worsening of macular edema [95,96,97] and hyphema [98,99].

In practice, adherence to topical eye drops is often overestimated and SLT relieves the burden of topical instillation, which is of concern especially in older people with comorbidities and low-self efficacy [100,101]. There is a reduction in SLT effect over time, with approximately 50% of failure after 2 years [102]. Most commonly, the success of SLT has been defined as IOP reduction from a baseline of at least 20% or of 3 mmHg or greater and not by achieving target IOP [103,104,105]. The recommendation for repeating SLT are required, because repeat SLT treatment is usually performed in clinical setting. Most of the studies evaluating retreatments with SLT after prior SLT or ALT were retrospective and performed in a small number of patients with medically uncontrolled glaucoma [106,107,108,109,110,111]. They reported that repeat SLT effectively lowered IOP in eyes with initial successful SLT [106,107,108,111] and controlled IOP up to 24 months in approximately 30% of eyes [108], whereas in one study [109] the effect of repeat SLT was half of the effect of the initial treatment.

The post hoc analysis of the LiGHT trial also investigated whether IOP-lowering efficacy and duration of effect of repeat SLT were comparable to initial SLT in medication-naïve open-angle glaucoma and ocular hypertensive eyes [112]. A total of 115 eyes of 90 patients received repeat SLT during the first 18 months of the trial. Absolute IOP reduction at 2 months was greater after initial SLT compared to repeat SLT, but when adjusting for pretreatment IOP (greater IOP in the initial SLT than in the repeat SLT), the absolute IOP reduction was greater in repeat SLT [112]. However, the comparison between retrospective studies and the LiGHT trial regarding the efficacy of repeat SLT is difficult due to different populations and criteria of success. At this moment, there are no randomized clinical trials to recommend how many times SLT can be repeated and is still effective. However, based on clinical experience, effect from SLT might be reduced after repeating it more than 2 or 3 times. As glaucoma is a lifelong disease, for the future, we need knowledge about long-term SLT outcomes such as visual field progression at 5- or 10-year follow up. 

## 6. Minimally Invasive Glaucoma Surgery

Minimally or microinvasive glaucoma surgery (MIGS) is a term collectively used to define a number of surgical procedures that involve a microincisional approach with minimal tissue trauma, have a higher safety profile than conventional drainage surgery and allow for rapid recovery with less impact on patients’ quality of life [113]. In recent years many of these surgical procedures have been implemented in clinical practice. MIGS procedures and devices lower IOP by draining aqueous into Schlemm canal, the subconjunctival space, or the suprachoroidal space [114]. By reducing the number of or the need for drops these procedures have the potential to reduce side effects of topical treatment and improve adherence [6].

MIGS procedures are bleb forming or non-bleb. In order to differentiate, the term microinvasive bleb surgery (MIBS) has been introduced. The distinction is important as bleb-forming procedures need meticulous post-operative management and experience in filtering surgery. To prevent scarring and establish flow, adherence to a topical anti-inflammatory treatment regimen is important. MIBS can be placed ab externo or ab interno (within the eye). Only those procedures with an ab interno approach with clear corneal incision and sparing of conjunctiva are considered MIGS [115,116].

The only MIBS with ab interno approach is XEN gel stent microshunt (Allergan, Dublin, Ireland). Meta-analysis of 78 studies found that XEN gel stent effectively reduced IOP and number of glaucoma drops till 48 months after surgery, but had a higher needling rate compared to trabeculectomy [117]. Three-year outcomes of ab interno XEN gel stand-alone procedure or combined phaco-XEN showed similar IOP-lowering from baseline and decrease in the number of eye drops (approximately halved) in both groups [118]. Needling over 3 years was required in 93 out of total 212 eyes (44%), with the mean number of needling of 1.3 per eye [118]. This suggests that careful post-operative follow up and interventions are important to maintain functioning bleb, all of which require patients’ adherence. Meta-analysis on the standalone XEN45 gel stent implantation in the treatment of open-angle glaucoma reported that the overall quality of current evidence is low, and there is the need for more randomized controlled trials and outcomes measured with a clinically meaningful definition of success [119].

MIGS implanted ab interno are required by the FDA to be performed at the time of cataract surgery. Therefore, the trial protocols compared a reduction in IOP and number of drops in eyes with MIGS combined with cataract surgery to cataract surgery alone [115]. MIGS implants that increase Schlemm canal outflow by either removing or bypassing juxtacanalicular trabecular meshwork tissue and inner wall of Schlemm canal include Trabectome^®^ (NeoMedix Corporation, Tustin, CA, USA), iStent inject (Glaukos Inc., San Clemente, CA, USA) and Hydrus microstent (Ivantis, Inc., Irvine, CA, USA). Although Trabectome was approved by the FDA in 2004 there has been only one randomized trial comparing ab interno removal of angle tissue with trabectome combined with cataract surgery to trabeculectomy combined with cataract surgery [120]. Phaco-trabectome achieved similar IOP lowering at 6 and 12 months compared to phaco-trabeculectomy with similar medications required at 1 year and no serious complications in the phaco-trabectome group. The trial has low quality evidence for the outcomes of ab interno trabectome surgery for open-angle glaucoma, with only 19 patients included and termination before the intended sample was reached [121]. Another procedure, also termed ab interno goniotomy or trabeculectomy, uses Kahook dual blade device (KDB, New World Medical Inc., Rancho Cucamonga, CA, USA) to excise and remove trabecular meshwork tissue, thus increasing aqueous outflow via Schlemm canal. In patients with mild to moderate glaucoma, adding ab interno trabeculectomy with Kahook dual blade to phacoemuslification was not more effective than phacoemulsification alone, with a similar safety profile [122]. Some studies found greater IOP reduction for goniotomy with Kahook dual blade [123,124]. However, the findings cannot be compared, as studies have different designs, study populations and criteria of success. It seems that this procedure modestly reduces IOP and the number of eye drops at 12 months, comparable to iStent [125,126].

Data from two RCTs suggested that iStent in combination with phacoemulsification was more effective in lowering IOP than phacoemulsification alone and reduced required daily topical medications by 0.4 drops more than cataract surgery alone at 1 year [127,128]. A greater reduction in IOP without medication for iStent inject (a second generation of iStent delivering 2 implants) combined with cataract surgery than for cataract surgery alone was sustained and present at 2 year-follow up [129]. The iStent inject trial captured patient reported outcomes using Visual Function Questionnaire 25 and Ocular Surface Disease Questionnaire [130]. The responses from both questionnaires suggest that reducing medication burden with iStent inject may improve quality of life by improving ocular surface symptoms and thus facilitating vision-related activities.

Hydrus microstent (Ivantis Inc., Irvine, CA, USA), an 8 mm intracanalicular scaffold device inserted through a corneal incision in the trabecular meshwork, lowers IOP by increasing aqueous outflow via Schlemm canal. It was approved by the FDA in 2018 with cataract surgery to treat mild to moderate glaucoma. A study comparing real-world 24-month outcomes for Hydrus (120 eyes) or iStent inject (224 eyes) combined with cataract surgery in patients with mild to moderate open-angle glaucoma or ocular hypertension showed sustained IOP reduction with a good safety profile and no significant difference in IOP reduction between the groups [131]. There may be a small additional reduction in glaucoma medication usage following cataract surgery with iStent inject compared to Hydrus [131]. Recent meta-analysis found moderate certainty evidence that adding a Hydrus safely improved the likelihood of drop-free glaucoma control at medium-term (6–18 months) and long-term (>18 months) follow up and conferred 2.0 mmHg greater IOP reduction at long-term follow up, compared with cataract surgery alone [132,133]. The latest systematic review and network meta-analysis including 6 prospective RCTs reported that the Hydrus implantation may have a slight advantage to achieve drop-free status versus the 1-iStent or 2-iStent implantation in combination with phacoemulsification [134]. Both device-augmented MIGS can reduce or delay the need for more invasive filtration surgery.

MIGS devices inserted ab interno that target the suprachoroidal route include the Cypass^®^ microstent (Alcon, Laboratories, Texas, Inc., Texas, USA), iStent SUPRA^®^ model G3 (Glaukos Inc., San Clemente, CA, USA) and the MINIject™ (iStar Medical, Wavre, Belgium). Cypass microstent combined with cataract surgery reduced IOP and the number of eye drops more than cataract surgery alone, but was associated with corneal endothelial cells loss and was withdrawn from the market by Alcon [135,136]. A study evaluating the efficacy and safety of suprachoroidal iStent SUPRA in conjunction with cataract surgery compared to cataract surgery alone has not published the results (NCT01461278). The MINIject, undergoing clinical trials, as a standalone procedure achieved 20% IOP reduction in all patients at 2 years [137], but a longer follow up to evaluate safety is required. MIGS devices using the suprachoridal pathway have not been a long-term success due to fibrosis and/or complications, hence improvement of material biocompatibility to limit foreign body reaction may overcome these barriers.

## 7. Discussion

Similarly to other chronic diseases, nonadherence to medication is a challenge to effective treatment of glaucoma. There are many barriers to adherence that need to be detected and the approach individually tailored [19]. The purpose of this review is to highlight different approaches to address adherence and relieve the burden of long-term frequent drop administration (Table 1). However, many of these approaches are still undergoing clinical trials.

Side effects of treatment are often reduced when switching from preserved to preservative-free eye drops and decreasing the number of daily drop instillations in patients with signs and symptoms of OSD. The toxic-inflammatory effects of BAK are well known and preservative-free drops should be a reference standard for all [138]. However, there is less information about the long-term influence of excipients on ocular surface. Freiberg et al. [139] observed that among preservative-free latanoprost products there were significant differences in pH value, osmolality, and surface tension which may lead to unstable tear film and ocular surface adverse effects. For the future, long-term clinical studies are required to evaluate the safety and efficacy of formulations with different physicochemical properties using a consensus-based series of outcomes and assessment methods [140].

In sustained drug delivery, nanotechnology-based treatments may have the potential to overcome the limitations of currently available glaucoma therapy as they enable targeted delivery, accurate dosing, less side effects, sustained release and increased bioavailability. Several glaucoma drugs have been investigated in nanomedicine formulation, but none of them is available for clinical use.

Of the sustained delivery systems, only the bimatoprost SR (Durysta^®^) intracameral implant has been approved for single administration due to safety issues until the results of the ongoing clinical trials on the long-term safety and efficacy of the implant are available. With the effect lasting up to 6 months, patients would need multiple administrations in aseptic conditions, which increases the risk of infections.

An important limitation of different drug delivery systems is that only one drug can be loaded, whereas the majority of glaucoma patients need more than one drug to achieve target IOP. Moreover, no studies on cost-effectiveness have been published, possibly because many of these new drug delivery systems are in the preclinical or clinical trials.

In the era of eHealth, smart drug delivery-connected devices in treatment of glaucoma have the potential to improve adherence and for the care provider to detect nonadherence and highlight the risk of progression. Smart drug delivery systems have been used only in research setting including small number of motivated subjects with a short follow up [141]. The size of these devices is still large, which may be inconvenient to adopt. Protection of data and sharing are important issues in digital health care and also fiduciary physician-patient relationship.

The SLT as first-line treatment was shown to be more cost effective compared to medication in the UK setting. It delays the need for eye drops in patients with OHT, early and moderate open-angle glaucoma. A retrospective study in a real-world setting found that 70% of patients initially responded to SLT, but approximately three-quarters of eyes failed treatment within 2 years post-SLT [142]. In this study, definition of failure included IOP ˃ 21 mmHg, IOP reduction < 20% from baseline and further glaucoma procedures (including repeat SLT) or medication increase from baseline.

Although there is increasing use of MIGS and MIBS devices and procedures, there is a lack of large randomized controlled trials and real-world observational studies to determine clinical and economic effectiveness. It is not clear whether the costs of using MIGS and MIBS are outweighed by the reduced number of medication and further intervention [143]. Cost-utility analysis using Markov model over lifetime horizon showed that iStent inject combined with cataract surgery is a cost-effective option for the treatment of patients with early to moderate glaucoma from the Italian NHS perspective [144]. MIGS may offer the advantage of a less rigorous follow up and post-op treatment regimen compared to bleb-forming procedures and some (Hydrus microstent, iStent inject) confer better IOP lowering and no or reduced medication need in early to moderate glaucoma compared to cataract surgery alone. 

Finally, identifying issues for poor adherence and addressing them in individual patient care using clear communication is critical [145]. A multifaceted approach including education, reminders, a regimen, and instillation techniques seems to work better in aiding adherence [65,72]. Future studies should focus on clinically important outcomes (e.g., VF progression), quality of life as well as the costs of intervention with longer post-interventional follow up.

## Figures and Tables

**Table 1 jcm-12-00151-t001:** Summary of treatment options to address low adherence.

Treatment Option		Type	Advantages	Disadvantages/Limitations	Clinical Setting
1. Medical treatment					
Topical PF drugs			Reduce signs and symptoms of OSD, thus may improve adherence	Drop instillation is not reduced	PF drugs available for most of the drug classes
Sustained DDS			No or reduced need of drop instillation depending on the DDS; reduction in systemic and local side effects	Depends on the DDS; Lack of data on the dosage and administration regimen of these formulations, metabolic ways and ocular toxicity of all formulation components, their pharmacokinetics and pharmacodynamics, the release of the drug in different eye tissues, formulation stability, the influence of the method of the synthesis not only on physio-chemical properties of formulation but also on its physiological effect, the suitability of nanocarriers with respect to biodegradability and patient comfort, safety issues	Only 1 sustained DDS approved for clinical use, single drug loading
	Nanoparticles	Different forms: liposomes, dendrimers	Improved corneal penetration, higher concentration at target tissue, longer retention, sustained release; different NP systems investigated carrying different glaucoma drugs	See above limitations	Not approved for clinical use; in preclinical and clinical trials
	Contact lenses	Various types of drugs or delivery systems can be placed into the periphery of lens	Increased drug residence time > 30 min improves bioavailability, prolonged drug release	Changes in contact lens swelling and water content, transmittance, protein adherence, surface roughness, tensile strength, ion, and oxygen permeability and leaching of the drug during contact lens manufacture and storage	Preclinical studies: contact lens eluting latanoprost starting human trial
	Extraocular inserts	Bimatoprost ring insert	As above, IOP -lowering effect over 6 months similar to timolol eye drops	Foreign body reaction to insert? Long-term acceptance -dislodgement. Cost	Not approved for clinical use
	Punctal DDS	Different pharmaceutical forms loaded into the core of the plug	Reduced need of drop instillation	Only low drug doses of potent drugs (e.g., prostaglandins) can be loaded into the plug matrix. Long-term acceptance. Side effects	Not approved for clinical use; in clinical trials
	Periocular DDS	Different DDS, subconjunctival injections	Reduced need of drop instillation	Efficacy and safety issues depending on the DDS	Most studies in animals. Not approved for clinical use
	Intraocular/intracameral drug delivery	Biodegradable implants using different DDS (bimatoprost, travoprost); titanium implant eluting travoprost (needs to be exchanged)	No or reduced need of drop instillation, effective IOP lowering ≥ 6 months	Retention of implant beyond the optimal drug effect, long-term safety, and repeatability	Bimatoprost SR intracameral biodegradable implant approved for single administration. Ongoing trials
2. Monitoring systems and smart DDS		MEMS caps for electronic monitoring. Smart drug delivery-connected devices	Supporting adherence by providing information to patients (alerts, remaining drug volume, positioning), health care providers	Studies showing improved adherence over short-term, measure drop dispensing and not drops landing into the eye. No real-world data, cost Data protection issues	Smart drug delivery-connected devices
3. SLT			Postpones the need for medical treatment, safe, can be repeated	Greater effect in eyes with higher pre-SLT IOP; reduction in effect over time. Lack of data about long-term SLT outcomes (visual field), how many times can be repeated and is still efficacious	As first treatment in open-angle glaucoma or high-risk ocular hypertension
4. MIGS		Microinvasive bleb surgery. Non-bleb forming	No drops or reduce the number of drops needed over 2 years; delay the need for more invasive surgery	Lack of large RCTs with long-term outcomes and real-world data on clinical and economic effectiveness	Available, combined with cataract surgery to treat mild to moderate glaucoma

DDS, drug delivery system; MIGS, minimally invasive glaucoma surgery; NP, nanoparticle; OSD, ocular surface disease; PF, preservative free; RCT, randomized controlled trial; SLT, selective laser trabeculoplasty.

## Data Availability

Not applicable.

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
