# Peer review of "Devices and Treatments to Address Low Adherence in Glaucoma Patients: A Narrative Review"

_jcm, 2022, doi:10.3390/jcm12010151_

Round 1
Reviewer 1 Report (Previous Reviewer 2)
I thank the authors for their careful consideration of my previous comments and suggestions, they have adequately addressed each of the points raised.
Author Response
Thank you for reviewing our manuscript and your comments.
Reviewer 2 Report (Previous Reviewer 1)
The authors did not address what they claim to be the main issue of the paper which is managing adherence. Again the review only summarize the various non medical treatment options, none of which aim to improve adherence bit rather provide alternatives to patients that are non-adherent.
Author Response
We acknowledge the reviewer’s comments, but humbly disagree. The review focuses on medical strategies, but also covers non-medical treatment options. Hence, the focus is on having a holistic view on each patient and to personalise treatments. This can be obtained by changing the medical choices, which we believe by itself can improve adherence, or by switching to laser and/or surgical strategies. We have highlighted this in the revised manuscript.
Reviewer 3 Report (New Reviewer)
Good review, although I would like a more extensive report to monitoring systems.
Author Response
Thank you for reviewing the manuscript. As suggested we have extend the report on monitoring systems/devices and highlighted the changes.
Reviewer 4 Report (New Reviewer)
This is a well-written and comprehensive review of devices and treatment options in patients with low adherence to glaucoma medical management. To make it easier for the reader to grasp the information, the authors should consider creating a summary table to summarize the various treatment options that they outline.
Author Response
Thank you for the positive review and suggestion to include a summary table.
We have added a table summarizing various treatment options.
Round 2
Reviewer 2 Report (Previous Reviewer 1)
While the interventions mentioned in this review might help improve adherence, the authors provide no evidence for such improvement in their manuscript. To the best of my knowledge, the issue of improving adherence hasn't been thoroughly evaluated in the literature in regard to such interventions (the main purpose of which is to have better control of IOP and not address the adherence issue). Therefore, the review in its current form is miss leading and uses the published data out of its original context
This manuscript is a resubmission of an earlier submission. The following is a list of the peer review reports and author responses from that submission.
Round 1
Reviewer 1 Report
The authors did an excellent job in reviewing both current and future glaucoma treatments.
However as the title suggests this was not the aim of the review. In the introduction the author clearly state that "The purpose of this review is to discuss new devices and drug delivery techniques (systems) as well as medical and laser treatments to improve adherence in patients with glaucoma." however the review does very little to describe the various methods and techniques that can be employed to improve adherence. If anything it lists various alternative treatments that can be used to avoid the need for adherence including long accting drug delivery systems, lasers and even surgery. None of which address adherence.
Author Response
REVIEWER 1
Comments and Suggestions for Authors
The authors did an excellent job in reviewing both current and future glaucoma treatments.
However, as the title suggests this was not the aim of the review. In the introduction the authors clearly state that "The purpose of this review is to discuss new devices and drug delivery techniques (systems) as well as medical and laser treatments to improve adherence in patients with glaucoma." however the review does very little to describe the various methods and techniques that can be employed to improve adherence. If anything it lists various alternative treatments that can be used to avoid the need for adherence including long acting drug delivery systems, lasers and even surgery. None of which address adherence.
Reply: Thank you for the comment. We agree that the title is misleading, and the review does not address methods and techniques to improve adherence. As suggested, we have changed the title to “Devices and Treatments to tackle Low Adherence in Glaucoma”. We have changed the last sentence of the Introduction that reads: “The purpose of this review is to discuss devices, drug delivery techniques (systems) and minimally invasive glaucoma surgery procedures as well as medical and laser treatments which can reduce the number of drops required to control intraocular pressure in patients with glaucoma.”
Reviewer 2 Report
This study provides a comprehensive review of the current medical and surgical treatment options available for glaucoma and outlines new treatments to improve adherence rates. I have just a few comments for improvement as follows:
1) There are a few minor grammatical and typographical errors in the manuscript, please proofread carefully to correct these errors. For example, on page 10 line 484, “refence” should be “reference”.
2) Page 6, lines 293-296. The authors state that visual field outcomes showed that a slightly greater number of patients treated with medical therapy first had fast visual field progression compared to those treated first with SLT. Can the authors report the actual numbers of patients here so that readers can see the exact difference in patient numbers as well as define what was considered “fast visual field progression” in the cited study?
3) The authors have addressed possible side effects of medication in detail but have not done the same for SLT. Can the authors please include potential complications and side effects that can arise from SLT and cite appropriate papers addressing this?
Author Response
REVIEWER 2
Comments and Suggestions for Authors
Overall, a very complete and thorough review. The title is a little misleading as MIGS procedures and SLT do not improve adherence but decrease medication requirements. A title along the lines: “Devices and Treatments to tackle Low Adherence in Glaucoma” may be more appropriate.
Reply: Thank you for the comment and suggesting a more appropriate title. We have changed the title to “Devices and Treatments to tackle Low Adherence in Glaucoma”.
Even though I am not a native speaker myself there are a few issues with language and grammar. I disagree with some tenses that were used and there are some issues with the use of the definite and indefinite article.
Reply: Thank you all comments. We have corrected all of them, as well as some typing and misspelling errors.
For example:
line 48: a significantly higher adherence
line 124: replace intraocular with intracameral
line 124: in different stages of development
line 131: add in size
line 143: drop instillations
line 146: regarding safety
line 153: device dislodgement (without article)
line 163: use plural
line 165: per device
line 158: the bimatoprost ring
line 272: hydrogel (spelling)
line 322: has been defined IOP reduction
line 356: bleb forming procedures (add procedures)
line 358: regimen (spelling)
line 365: replace solo with stand-alone
line 375: concomitant is redundant
line 380: capitalize Trabectome
line 393: with a similar safety profile
The manuscript could benefit from a review by a native speaker and would improve significantly in clarity.
There are minor issues regarding style.
Some products are mentioned both with their trade names, manufacturing company and substances where others only by commercial name or only by substance i.e line 205 Ozurdex. Trade names and manufacturing company should be mentioned in their first appearance in the text. This is not the case for the bimatoprost implant.
Reply: Added trade name and manufacturing company
The paragraphs on smart delivery systems appear to be out of order and should be mentioned before SLT and MIGS devices.
Reply: The paragraphs on smart delivery systems have been inserted before SLT, and these order has been introduced throughout the manuscript (including abstract).
Content:
line 18: there are also MIGS procedures without implants
Reply: Thank you, everything corrected
line 54: I believe that in the CITGS study adherence was not accessed on the day of follow-up but through telephonic interviews on different days. This is a strength of the study as non-adherence is often not admitted in front of the treating physician.
Reply: Thank you for the comment. We have changed to: “In another study, patients randomized to the treatment arm of the Collaborative Initial Glaucoma Treatment Study were assessed for medication adherence using telephone interviews scheduled at the same 6-month intervals as clinical visits but on different days and followed up for an average of 7.3 years. Scheduling timing of telephone interviews independently of clinical visit represents a strength of the study as non-adherence is often not admitted in front of the treating physician.
line 95: replace frequency with prevalence
Reply: Thank you, replaced.
line 103: consider adding fixed combination of timolol with dorzolamide which is also available in preservative free formulation
Reply: Thank you, we have added preservative-free fixed combination of timolol/dorzolamide.
line 173: you may want to comment on the superb efficacy of timolol which is likely the effect of decreasing wash-out through the nasolacrimal ducts by inert punctal plugs
Reply: Thank you for your comment. We have added the sentence: “Superb efficacy of timolol eye drops is likely the effect of decreased wash-out through the nasolacrimal ducts by inert punctal plugs.”
line 295: cost effectiveness studies are pertinent to the country the study was conducted
Reply: Thank you. We have added to the sentence: Cost-effectiveness analysis, pertinent to the United Kingdom where the trial was conducted, showed…
line 323: this sentence is unclear
Reply: We have rephrased the sentence for better understanding and divided it into 2 shorter sentences: “Most of the studies evaluating retreatments with SLT after prior SLT or ALT were retrospective and included a small number of patients with medically uncontrolled glaucoma (ref). They reported that repeat SLT effectively lowered IOP in eyes with initial successful SLT (Hong 2009; Avery 2013; Khouri 2014; Polat 2016; Ilveskoski 2019) and controlled IOP up to 24 months in approximately 30% of eyes (Khouri 2014), whereas in one study (Hutnik 2019) the effect of repeat SLT was half of the effect of the initial treatment.”
line 340: this statement contradicts an earlier one according to which repeat SLT (2nd procedure) is more effective than initial SLT.
Reply: The line numbers are different in our final version of submitted manuscript. We assume it is a sentence starting with “But based on clinical experience…
We do not think this sentence is contradictory to the earlier statements including post-hoc analysis of the LIGHT trial reporting data of repeat SLT at 2 months. The LIGHT trial included treatment-naïve patients and success was defined as reducing IOP to target or beyond (different from the populations and success criteria of other retrospective studies evaluating efficacy of repeat SLT). In the study, adjusted absolute IOP reduction at 2 months (adjusting for pretreatment IOP before initial or repeat SLT) was greater after repeat SLT than initial SLT.
For improved clarity we have changed to: “Absolute IOP reduction at 2 months was greater after initial SLT compared to repeat SLT, but when adjusting for pre-treatment IOP (greater IOP in the initial than in the repeat SLT), the absolute IOP reduction was greater in repeat SLT. However, the comparison between retrospective studies and the LIGHT trial regarding the efficacy of repeat SLT is difficult due to different populations and criteria of success. At this moment, there are no randomized clinical trials to recommend how many times SLT can be repeated and is still effective. But based on clinical experience, effect from SLT might be reduced after repeating it more than 2 or 3 times.”
line 363: it is hard to explain why phaco XEN had a lower needling rate compared to XEN alone. I would stick to the comparison with trabeculectomy only.
Reply: As suggested, we have left out the comparison phaco Xen to Xen alone.
line 394: There are larger studies that claim higher IOP reduction for KDB i.e. Elmallah MK et al, Iwasaki K et al.
Reply: The retrospective study of of ElMallah MK et al. included 42 eyes of 35 patients (the same number as a prospective RCT of Ventura-Abreu) with medically uncontrolled glaucoma undergoing stand-alone excisional goniotomy using KDB. Authors reported significant IOP reductions (mean baseline IOP 21.6 mmHg) averaging 4–6 mmHg through 12 months of follow-up with significant reduction in medication from baseline of 19-28% through 6 months after surgery, but the difference in the number of medication from baseline was no more significant at 12 months. Interestingly, 5 out of 6 patients requiring additional glaucoma surgery (trabeculectomy and Ahmed drainage implant) had higher baseline IOP.
The study of Iwasaki K et al. retrospectively evaluated long-term outcomes of phacoemulsification combined with KDB goniotomy in 148 eyes of 97 patients with open-angle glaucoma. Surgical failure was defined with or without medication as a <20% reduction in preoperative IOP or IOP > 18 mmHg (criterion A), IOP > 14 mmHg (criterion B), or requirement for reoperation (a complicated definition). They found that the probability of success at 36 months was 52,5% and 36.9% using a more stringent criterion.
As studies have different designs (prospective RCT, retrospective; stand-alone vs combined procedures), populations and criteria of success their findings cannot be compared.
We have commented as follows. “Some studies found greater IOP reduction for goniotomy with Kahook dual blade ( ElMallah MK et al.; Iwasaki K et al.). However, the findings cannot be compared, as studies have different designs, study populations and criteria of success.”
Line 394: I also don’t understand why KDB and i-Stent are the only procedures compared. There are for example excellent data on i-Stent vs. Hydrus.
Reply: Comparison Istent vs Hydrus: as suggested we have included two recent studies Holmes DP et al. Clin Exp Ophthalmol 2022 and a recent meta-analysis comparing Hydrus vs iStent (Hu R, et al. BMJ Open 2022)
“A study comparing real-world 24-month outcomes for Hydrus (120 eyes) or iStent inject (224 eyes) combined with cataract surgery in patients with mild to moderate open angle glaucoma or ocular hypertension showed sustained IOP reduction with a good safety profile and no significant difference in IOP reduction between the groups. There may be a small additional reduction in glaucoma medication usage following cataract surgery with iStent inject compared to Hydrus (ref. Holmes DP, et al. 2022).”
“Just recently published systematic review and network meta-analysis including 6 prospective RCTs reported that the Hydrus implantation may have a slight advantage to achieve drop-free status versus the 1-iStent or 2-iStent implantation in combination with phacoemulsification (Hu K, et al 2022)”.
line 411: reduction in the rate of patients requiring filtration surgery is also important in my opinion
We have added this sentence and reads as follows: “Both device-augmented MIGS can reduce or the delay the need for more invasive filtration surgery”.
Reviewer 3 Report
Overall, a very complete and thorough review. The title is a little misleading as MIGS procedures and SLT do not improve adherence but decrease medication requirements. A title along the lines: “Devices and Treatments to tackle Low Adherence in Glaucoma” may be more appropriate.
Even though I am not a native speaker myself there are a few issues with language and grammar. I disagree with some tenses that were used and there are some issues with the use of the definite and indefinite article.
For example:
line 48: a significantly higher adherence
line 124: replace intraocular with intracameral
line 124: in different stages of development
line 131: add in size
line 143: drop instillations
line 146: regarding safety
line 153: device dislodgement (without article)
line 163: use plural
line 165: per device
line 158: the bimatoprost ring
line 272: hydrogel (spelling)
line 322: has been defined IOP reduction
line 356: bleb forming procedures (add procedures)
line 358: regimen (spelling)
line 365: replace solo with stand-alone
line 375: concomitant is redundant
line 380: capitalize Trabectome
line 393: with a similar safety profile
The manuscript could benefit from a review by a native speaker and would improve significantly in clarity.
There are minor issues regarding style.
Some products are mentioned both with their trade names, manufacturing company and substances where others only by commercial name or only by substance i.e line 205 Ozurdex. Trade names and manufacturing company should be mentioned in their first appearance in the text. This is not the case for the bimatoprost implant.
The paragraphs on smart delivery systems appear to be out of order and should be mentioned before SLT and MIGS devices.
Content:
line 18: there are also MIGS procedures without implants
line 54: I believe that in the CITGS study adherence was not accessed on the day of follow-up but through telephonic interviews on different days. This is a strength of the study as non-adherence is often not admitted in front of the treating physician.
line 95: replace frequency with prevalence
line 103: consider adding fixed combination of timolol with dorzolamide which is also available in preservative free formulation
line 173: you may want to comment on the superb efficacy of timolol which is likely the effect of decreasing wash-out through the nasolacrimal ducts by inert punctal plugs
line 295: cost effectiveness studies are pertinent to the country the study was conducted
line 323: this sentence is unclear
line 340: this statement contradicts an earlier one according to which repeat SLT (2nd procedure) is more effective than initial SLT
line 363: it is hard to explain why phaco XEN had a lower needling rate compared to XEN alone. I would stick to the comparison with trabeculectomy only.
line 394: There are larger studies that claim higher IOP reduction for KDB i.e. Elmallah MK et al, Iwasaki K et al. I also don’t understand why KDB and i-Stent are the only procedures compared. There are for example excellent data on i-Stent vs. Hydrus.
line 411: reduction in the rate of patients requiring filtration surgery is also important in my opinion
Author Response
We have received comments from 2 reviewers.
Round 2
Reviewer 1 Report
The review in its current form does not have clearly defined purpose and is very disorganized in general. Not only is there nothing novel in the way the authors chose to present the information, which can be found more easily in previous publications, but the mixture of topics make it confusing. I personally found it hard to understand what was the message
Author Response
What was in the first review an excellent job in reviewing both current and future glaucoma treatments (and the main comment was inappropriate title that we have changed based on the suggestion of the Reviewer 2), in the second round you stated that “the review has no clearly defined purpose and is very disorganized in general… no novel information, which can be found more easily in previous publications.” Authors, both glaucoma specialists, believe that the topic is of interest, being a comprehensive overview of treatments that may help to improve or reduce the need of adherence, which is an important issue in glaucoma treatment.